# Exploring the Potential of Microbial Coalbed Methane for Sustainable Energy Development

**DOI:** 10.3390/molecules29153494

**Published:** 2024-07-25

**Authors:** Yu Niu, Zhiqian Wang, Yingying Xiong, Yuqi Wang, Lin Chai, Congxiu Guo

**Affiliations:** 1School of Electric Power, Civil Engineering and Architecture, Shanxi University, Taiyuan 030006, China; 202323505024@email.sxu.edu.cn (Z.W.); xiongyy@sxu.edu.cn (Y.X.); 202323505023@email.sxu.edu.cn (Y.W.); guocongxiu@sxu.edu.cn (C.G.); 2Beijing Key Laboratory of Green Chemical Reaction Engineering and Technology, Department of Chemical Engineering, Tsinghua University, Beijing 100084, China; chailin@mail.tsinghua.edu.cn

**Keywords:** microbial transformation, biodegradation of coal, biomethane, MECoM, sustainable energy development

## Abstract

By allowing coal to be converted by microorganisms into products like methane, hydrogen, methanol, ethanol, and other products, current coal deposits can be used effectively, cleanly, and sustainably. The intricacies of in situ microbial coal degradation must be understood in order to develop innovative energy production strategies and economically viable industrial microbial mining. This review covers various forms of conversion (such as the use of MECoM, which converts coal into hydrogen), stresses, and in situ use. There is ongoing discussion regarding the effectiveness of field-scale pilot testing when translated to commercial production. Assessing the applicability and long-term viability of MECoM technology will require addressing these knowledge gaps. Developing suitable nutrition plans and utilizing lab-generated data in the field are examples of this. Also, we recommend directions for future study to maximize methane production from coal. Microbial coal conversion technology needs to be successful in order to be resolved and to be a viable, sustainable energy source.

## 1. Introduction

Coal is one of the major fossil fuels. It has a wide range of uses as a fossil fuel, especially in the power, steel, and chemical industries. As a result, the demand for coal is increasing. Data from the International Energy Agency (IEA) indicate that international coal consumption hit another record high in 2023. As shown in Figure 1, coal consumption has steadily risen from 4681 Mt in 2000 to 7997 Mt in 2013, rising again after reaching a first low in 2016, before reaching a second peak of 7833 Mt in 2018 and a second low in 2020, but it is expected to continue to increase in the following years. These data visualize the huge demand for coal. However, in the process of mining and utilizing coal in large quantities, many environmental problems have arisen, posing a significant challenge to its sustainable development, such as surface subsidence, dust pollution, groundwater pollution, and the release of harmful gasses from combustion. Seeking a more scientific way of development is a necessary way for the sustainable development of coal.

Microbial-enhanced coalbed methane (MECoM) technology presents a sustainable solution to solve several problems arising from increasing coal consumption. This technology involves a natural process that mimics and enhances the biogenicity of coal gas to achieve sustainable use of the resource. Coal microbiology, as a burgeoning field, holds significant importance in the realms of environmental conservation and the advancement of the coal industry. Presently, there exists considerable potential for leveraging this discipline to enhance coal utilization efficiencies, mitigate pollution, and foster the emergence of novel energy reservoirs. Through mechanisms such as biodegradation, bioremediation, and reclamation [1,2,3,4,5,6], microbes exhibit the capacity to convert coal into water-soluble compounds or hydrocarbon gasses, which can subsequently be harnessed for the production of clean chemicals and fuels [7,8,9,10]. The utilization of microbial digestion technology stands as a valuable strategy in this regard. Microbial digestion technology represents a valuable approach for the clean coal industry since it can convert organic components in coal into chemical substances or clean fuels. Despite its enormous potential, extensive research is necessary to optimize the specificity and efficiency of microbial degradation, as well as to adapt this technology to industrial production.

For this technology with great potential, scholars have conducted a lot of research on strains, pretreatments, mechanisms of action, and influencing factors. Cohen [11] found that there are two types of fungi that can break down coal. This result emphasizes how culture factors have a major influence on how efficiently coal degrades [12,13]. By manipulating these conditions, such as through the addition of organic nitrogen sources like maltose, we can influence the behavior of certain strains [14]. It is noteworthy that a particular local strain may possess the potential to degrade coal [15]. Pretreatment can alter the nature of coal, making it more amenable to biodegradation. Hydrogen peroxide pretreatment of sub-bituminous coals, in particular, holds great promise [16]. Experimental data support the degradability of both coal and mudstone samples—the maximum biogenic methane yields for coal and mudstone, respectively, are 98.5 and 72.5 μmol/g [17]. Yang’s latest research [18] effectively replicated the production of methane in a 160 L fermenter with lump anthracite amendment. This enrichment produced a maximum methane production of 13.66 μmol/mL and roughly 218.56 μmol CH_4_/g of coal. This offers valuable insights into the potential methods for converting coal resources into methane.

Scott’s perspective [19] is that converting just 1% of US coal resources to methane could increase natural gas resources by 23 trillion cubic feet. This conversion process is feasible due to the laccase activity exhibited by certain microorganisms [20].

A vast number of microbial populations work cooperatively to degrade the methane produced through a complex mechanism. Despite the fact that certain microorganisms do not directly produce methane, their byproducts can be extremely important for cleaning coal [21]. Moreover, during the coal biogasification process, carbon conversion can be improved, nitrogen and sulfur can be fixed, and dehydrogenation and deoxygenation reactions can occur. Research on turning coal into valuable resources can proceed now that this knowledge has been established [22].

Microbial degradation of coal also plays an important role in managing environmental pollution. Runnion and Combie [23] used biological samples from Yellowstone to reduce the contents of pyrite and sulfate sulfur by 90% and organic sulfur by 33% in North Dakota lignite. Caseous roots in microbial degradation of coked soils dominated by coal and coal tar may be a potential carbon source for microorganisms [24]. Coal gasification wastewater (CGW) can be degraded anaerobically more quickly by converting waste sludge into nitrogen-doped sludge carbon (N-SC), a unique conductive material [25]. Nitrogen-doped sludge-activated carbon (Fe_3_O_4_/NSBAC) loaded with Fe_3_O_4_ can be used to improve the degradation efficiency of coal gangue effluent [26]. 

The number of microorganisms is inversely correlated with PAH concentration, and PAH bioavailability increases with increasing soluble organic matter concentration [4]. Remedial sites of mining contamination can also benefit from tree planting. For individual tree species, total PAH reduction decreased in the following order: *C. siamea* (81.6%) > *A. lebbeck* (55.6%) > *D. regia* (51.9%) > *D. sissoo* (51.5%) [27]. Rich microbial diversity that encourages the activity of degrading bacteria has been created by the action of these trees.

The process of coal biodegradation necessitates the cooperation of hydrolyzing and methanogenic bacteria. However, it is noteworthy that the kinetics of these biological processes are consistently slower compared to those of thermochemical processes, which is unsurprising. Additionally, the yields obtained are lower as well [28]. Subsequently, enhancing the biodegradation efficiency and profitability of coal through artificial interventions assumes precedence in our research efforts. 

As coal microbiology encompasses several topics and is an interdisciplinary field of study, the studies of the above authors are not sufficient to identify research hotspots for MECoM. To investigate the links between the different fields, we searched Web of Science from 1990 to 2024 for ‘coal’, ‘microbiology’, and ‘degradation’, finding 666 papers. Using the co-occurrence feature in the VOSviewer 1.6.20, the following important research hotspots and trends were also identified, as shown in Figure 2. We selected 78 keywords with a frequency of 15 or more occurrences in the 666 articles for extraction. We can observe that the research trend revolves primarily around the theme of ‘degradation’, with ‘biodegradation’, ‘coal’, and ‘microbial communities’ being the key areas of focus that have been extensively studied. Among them, ‘coal’, ‘microbial communities’, ‘bacterial strains’, ‘sewage’, ‘sludge’, ‘biodegradation’, etc., are the main research hotspots. The current focus of research is indicated by these hotspots. To further understand the traits and roles of coal microbial communities, more integrated studies are required because of the intricate connections between the research themes [29]. To fully explore the potential and applications of coal microbiology, we must thus conduct deeper research in ecology and biotechnology [30]. Coal microbiology is expected to contribute more to the growth of the coal industry and environmental protection with further research and investigation, in our opinion.

This review aims to provide a comprehensive overview of microbial-enhanced coalbed methane (MECoM) production. It delves into the degradation process of MECoM, offering a detailed summation of the microbial species involved. Furthermore, it summarizes and analyzes the environmental factors that influence microbial degradation, specifically tying them to practical use cases. Additionally, this paper presents an outlook on the future of MECoM, accompanied by expressed opinions on its potential. After an extensive examination of numerous references, it is conclusively stated that MECoM holds significant promise as a sustainable and renewable energy source.

## 2. Microbially Coal Degradation Pathways and Methane Formation

### 2.1. Overview of the Degradation Process

As thorough reviews have demonstrated, a variety of bacteria and their interactions play a part in the multi-step microbial degradation of coal to produce biomethane. The products from the decomposition of the previous stage of the microbial community can be used as substrates for the next stage of the microbial community, thus forming a continuous chain of degradation, as shown in Figure 3. Through a number of enzyme-based processes, including hydrolysis, fermentation, acid production, hydrogen generation, and methane production, microorganisms are able to produce methane from coal [31]. Microorganisms degrade coal’s complex organic content into simple components, creating fatty acids, alcohols, and hydrogen [32,33,34,35]. During further depolymerization and hydrolysis, the fatty acids are hydrolyzed into smaller organic molecules [36,37]. The methanogenic bacteria then break down these small organic molecules into methane and hydrogen. Temperature, humidity, and pH are all essential factors in this process [38,39]. Both the chemical content and the structure of the coal influence the breakdown process. The microbes thus degrade the coal into coalbed methane, increasing the utilization of the coal. 

The equations for the three processes of hydrolysis, fermentation, and hydrogen and acetic acid production are as follows:

Hydrolysis:$$R-X+H_{2}O\to R-\mathrm{OH}+X^{-}+H^{+}$$

Fermentation:

Fatty acids first undergo activation by combining with CoA to form acyl-CoA, which then enters the β-oxidation cycle in microorganisms. Within the β-oxidation cycle, acyl-CoA undergoes four sequential steps: dehydrogenation, hydration, re-dehydrogenation, and thiolysis. These steps ultimately result in the gradual breakdown of fatty acids into acetyl-CoA, accompanied by a release of significant energy, primarily in the form of ATP, for utilization by the microorganism. In this process, it can provide microorganisms with the energy they need to survive, as well as breaking down long-chain fatty acids into shorter chains. The process is shown in Figure 4.

Hydrogen and acetic acid production:$${\mathrm{CH}}_{3}{\mathrm{CH}}_{2}\mathrm{OH}+H_{2}O\to {\mathrm{CH}}_{3}\mathrm{COOH}+2H_{2}$$
$${\mathrm{CH}}_{3}{\mathrm{CH}}_{2}\mathrm{COOH}+2H_{2}O\to {\mathrm{CH}}_{3}\mathrm{COOH}+3H_{2}+{\mathrm{CO}}_{2}$$

The permeability of the coal, the porosity through which the microorganisms can pass, the nutrients required by the microorganisms, etc., also affect biogas production [40,41,42]. 

Additionally, coal rank may have an effect on microbial degradation. Numerous studies in the literature have shown that anthracite is more difficult to degrade compared to sub-bituminous coal and lignite because anthracite is more structurally stable and has a higher degree of variability [43,44,45,46,47,48,49]. This makes anthracite require longer hydrolysis times as well as longer gas production times, resulting in weaker degradation than lignite. However, Fallgren [50] believes that methane production from coals with a higher rank is higher than that from coals with a lower rank. According to Wawrik’s research, the amount of methane produced by the microbial degradation of coal is independent of the coal’s rank [39]. The quality of the coal is not the only factor that affects the microbial production of methane [51]. The volatiles in anthracite coal provide another source of feedstock for the production of methane [52]. Biomethane production was found to positively correlate with volatile matter content, H, N, and coal rank, according to Bao’s research [43]. The distribution of biomethane production was also regular. There was an inverse relationship between S content and the biomethane yield and inert matter content. Coal seams have a lot of pores and fissures, as shown in Figure 5, which allow methanogenic bacteria and hydrolysis to grow unrestricted and break down the coal through groundwater flow and self-reproduction. Nutrient solution injection, artificial bacterial liquid, fracturing fluid, etc., can quicken the pace at which microorganisms penetrate and spread.

When arriving at the methane production process through multiple fermentation and degradation in the coal seam fissures, the microorganisms complete the methane formation and production in three ways [53]. These three methane production methods and formulas are as follows:

Hydrogenotrophic methanogenesis: microorganisms produce methane using hydrogen and carbon dioxide as substrates [42,54,55,56,57]:$${\mathrm{CO}}_{2}+4H_{2}\to {\mathrm{CH}}_{4}+2H_{2}O$$

Acetic acid fermentation: microorganisms produce methane by breaking down acetic acid [58,59]:$${\mathrm{CH}}_{3}{\mathrm{COO}}^{-}+H^{+}\to {\mathrm{CH}}_{4}+{\mathrm{CO}}_{2}$$

Methylotrophic methanogenesis: microorganisms produce methane through methyl compounds:$${\mathrm{CH}}_{3}\mathrm{OH}+2H^{+}\to {\mathrm{CH}}_{4}+H_{2}O$$

Typically, both land and water bodies contain this process of producing methane [60]. Of course, the biomethane types all depend on certain microbial communities and suitable environmental conditions [57]. The emergence of heavy hydrocarbons across the entire biodegradation process explains why methane is mostly formed through the fermentation of fatty acids rather than through bioconversion from carbon dioxide [61]. 

In summary, coal biodegradation is a complex process that involves a variety of microbes and reaction stages. The process is also influenced by environmental variables such as pH, temperature, and nutrient availability [62,63], which mainly affect the kinds of flora, microbial development, enzyme activity, metabolic pathways, and other associated aspects [60,61]. Understanding these effects can help with predicting and controlling coal biodegradation.

### 2.2. Microbial Communities Involved in Coal Degradation

Coal reserves usually contain continuous, active microbial populations, as shown by studies carried out in situ and with laboratory cultures [64]. Having compared the gas sample to other samples, we found a noticeable decrease in microbial diversity. Depending on regional variables and the features of the coal reservoir, the microbial community involved in coal degradation has a variety of properties. Water sources are known to carry nutrients and electron acceptors that advance methanogenesis in coalbed methane (CBM) reservoirs, in addition to introducing microbial communities into coal reservoirs. Proteobacteria, Firmicutes, and Bacteroidetes bacilli are frequent visitors to coal seam methane (CSM) reservoirs [65]. Actinomycetes, fungi, and bacteria make up the majority of the microbial community during coal degradation [32,66]. It is crucial to remember that experimental precision and other factors can leave additional bacterial populations unidentified. A variety of microorganisms are present in coal seam and coalbed water, the bacteria having a higher abundance than archaea. A coal seam microbial’s structure has previously been thoroughly investigated utilizing polymerase chain reaction (PCR)-acquired clone libraries containing incomplete 16s ribosomal RNA genes. All of the coals that were examined had small-subunit rRNA sequences that were identified as belonging to the aerobic, facultatively anaerobic, and anaerobic genera. A wide range of species, including fungi, fermenters, acetogens, heterotrophic bacteria, and syntrophs, may catalyze these processes. Methanogenic bacteria are divided into four main parts: Methanobacteria, Methanococci, Methanomicrobia, and Methanopyri [67,68]. These 4 classes include 7 orders, 14 families, and 35 genera. The species, names, and functions of various microorganisms implicated in coal degradation that have been described in earlier studies are briefly summarized in Table 1. The microbial community patterns of different coal seams vary accordingly [69]. Actinobacteria, Mycobacteria, and Phylum Firmicutes made up the majority of the bacterial population, whereas Methanogenus makes up the majority of the archaeal group. Bacillus, Pseudomonas aeruginosa, and Streptomyces can break down organic matter into ethanol, acetic acid, lactic acid, carbon dioxide, and hydrogen [70]. These bacteria are important in the breakdown of coal. The phylum Firmicutes mainly produces intermediates such as acids and alcohols [64]. Furthermore, fungi that can grow in a variety of environments are essential to the breakdown of coal. Studies have shown that the quantities of volatile fatty acids and rather stable fungal populations are favorable for hydrogenotrophic methanogenesis [71,72]. Aspergillus, Penicillium, and Xylomycetes species aid in the decomposition of coal [64]. Methanogenic archaea are a unique family of microbes that exhibit amazing survival and conversion skills in the constrained anaerobic coal seam habitat. Through the skillful conversion of basic molecules like hydrogen, carbon dioxide, formate, or acetate into methane, they are able to receive the energy they require to thrive. Iron-reducing bacteria (IRB) and sulfate-reducing bacteria (SRB) compete with methanogenic archaea for the use of acetate. In one study, only 28% of the amended acetate accounted for methane production; the majority was consumed by SRB and IRB [73].

The makeup, roles, and metabolic processes of the microbial communities are frequently unique to a coal basin and can even differ depending on where the basin is located. Similarly, environmental conditions such as temperature, pH, and nutrient availability can also influence the microbial communities involved in coal degradation. Aromatic compounds in some coals also inhibit microorganisms, causing changes in microbial communities. 

Furthermore, stringent anaerobes that are involved in methanogenesis can be found in a variety of subterranean habitats. Research on the makeup of the microbial communities in CBM reservoirs has revealed a variety of bacterial and archaeal assemblages. Most of the interactions are involved in obligate interdependence. There are two types of interactions in the metabolism of methanogens and non-methanogenic bacteria: interactions with fermentation bacteria and interactions with obligate H_2_ production of acetic acid bacteria. Aerobic microorganisms exhibit a higher propensity for the degradation of heterocycles in coal, while anaerobic microorganisms employ the release of extracellular enzymes to facilitate the production of methane gas [46]. 

In laboratory microcosm investigations, organisms belonging to the phylum Firmicutes, which includes acetogens and fermenters, may predominate even though they are typically a minor part of the in situ microbial community. Although additives provided to microorganisms to promote methanogenesis may enhance fermentation in laboratory experiments, they may still form an important part of the methanogenic community.

Despite tremendous advancements in lab research to speed up the process of producing biogenic methane, the field has not yet achieved increased production of CBM. Therefore, discovering and finding more strains is one of our main directions. It is feasible to find efficient, stable, and adaptable strains to achieve the efficient application of MECoM through domestication in the laboratory.

### 2.3. Enzymes and Metabolic Pathways Involved in Coal Degradation

Enzyme–metabolite interactions take place throughout the intricate process of microbial coal breakdown. Coal is a multifaceted, intricate biological substance. Microbial enzymes are capable of dissolving coal into its constituent parts, which include cellulose, carbohydrates, phenolic compounds, and lignin [100]. Microorganisms create enzymes that are essential for the depolymerization and hydrolysis of coal’s organic components. Coal is broken down by enzymes including cellobiohydrolases, xylanases, laccases, lignin peroxidases, and manganese peroxidases [101,102]. Lignin is enzymatically degraded into aromatic compounds for metabolic degradation by other microorganisms [7,72,88]. The breakdown of cellulose and hemicellulose, two important components of coal, is facilitated by the enzymes cellulases and hemicellulases [103]. These enzymes are secreted by microorganisms such as fungi and bacteria for degrading coal macromolecules [43,69]. The specific enzymes and metabolic pathways that coal goes through to break down can vary depending on the microbial populations and environmental factors. Numerous enzymes generated by microbes, such as peroxidases, phenol oxidases, hydrolases, glucose oxidase, amylase, and others, facilitate the depolymerization process [7]. Microbes can metabolize sugars like glucose through enzymatic mechanisms, making them available for further use. Along with other compounds like pyruvate, amounts of ATP and NADH are created during the enzymatic breakdown of glucose. These compounds function as both energy sources and substrates for a variety of metabolic activities. Apart from fermentation, other metabolic pathways like the tricarboxylic acid cycle (TCA cycle) and the electron transport chain (ETC) are necessary for the production of ATP and the energy molecules NADH and FADH2. Depending on the type of coal and microbe, different metabolic pathways are involved in coal degradation. For instance, lignin and cellulose prese can be broken down by fungi like Aspergillus and Penicillium, while bacteria like Rhodococcus and Mycobacterium can break down the aliphatic and aromatic hydrocarbons found in coal. The degradation products of coal can be further metabolized by microorganisms to produce useful products such as methane, ethanol, and hydrogen [103].

The acquisition of products and energy that act as substrates and sources of energy for other metabolic processes depends on these mechanisms. In the methanogenesis process, there are three pathways through which substrates are consumed and ATP is produced. The ATP yield per 1 mol of substrate consumed follows a specific order: hydrogen-supporting yield > methyl-supporting yield > acid-supporting yield. It is important to note that this sequence determines the efficiency of energy production in methanogenic microorganisms. Thus, for the purpose of maximizing biogas production and other related applications, it is imperative to comprehend the mechanisms underlying each pathway and their respective energy yields. Enzymes are extremely selective substrate catalyzers that are recyclable when used under favorable conditions, with catalytic rates up to 10^6^–10^20^ times higher than those of typical non-biological catalysts, and can be reused [100]. 

Microorganisms act in a variety of ways, and the processes described above are far from sufficient; a clear understanding of the mechanisms of action of microorganisms requires a variety of instrumentation.

Coal biodegradation is usually studied using known enzymes and various characterization methods such as strain gene sequencing, scanning electron microscopy, infrared spectroscopy, nuclear magnetic resonance (NMR), and liquid/gas chromatography. The impact of a single, unidentified enzyme on coal decomposition has been demonstrated through the use of enzyme extraction and purification techniques. One example is the isolation and purification of Penicillium decumbens P6’s esterase using a series of techniques including ammonium sulfate precipitation, anion exchange, and gel filtration chromatography. The crude esterase was compared to the purified esterase, and the latter was analyzed for its ability to depolymerize lignite coal.

Enzymes generated by bacterial strains are secreted outside the cell during coal breakdown. The functions, quantitative variations, expression profiles, and interaction mechanisms of these enzymes remain largely unknown. Understanding the roles played by biological enzymes in the breakdown of coal requires knowledge of this information. To gain a comprehensive and in-depth understanding of the complex activities of biodegradation of coal, proteins must be studied at holistic, dynamic, and network levels. The rapid development of proteomics provides an answer to this problem. Proteomics is the science of studying the composition of proteins and their changing patterns at the level of cells, tissues, or organisms as a whole, taking the proteome as the object of study. The aim of this study is to analyze the various attributes of proteins encoded by the genome that play a role in biological processes, such as protein expression levels and post-translational modifications.

Presently, quantitative proteomics methods utilizing liquid chromatography–tandem mass spectrometry (LC-MS/MS) are widely used. Among these techniques, the stable isotope-labeling TMT method has gained popularity as a reliable method for relative quantification in proteomics research due to its low quantification error, high sample throughput, and sensitivity in protein detection.

## 3. Applications of Microbial Coal Degradation for Sustainable Energy Development

Using microorganisms to break down coal for chemical synthesis and sustainable energy production is a novel approach known as microbial coal degradation technology. It offers a novel approach to the efficient use of coal resources and the conversion of clean energy, and it has a wide range of applications in the development of sustainable energy.

### 3.1. Coal Bioconversion for Energy Production

Applications of microbial coal-degrading technology in energy production are significant. Coal may be transformed by microorganisms into sustainable energy sources including ethanol, methanol, and methane. The primary ingredient of natural gas, methane, is a clean, effective energy source. Microorganisms can partially replace conventional fossil fuels and minimize contaminated gas emissions by converting coal to methane, which offers the benefits of low energy consumption and environmental protection. The technologies employed are categorized into four main areas: microbial production enhancement through in situ nutrient modification, bio-enhancement via the injection of enriched cultures, the enhancement of coal recoverability through physical fracturing, and the enhancement of bioavailability through biotic or abiotic pretreatment [104,105]. These four methods are applied as shown in Table 2. Large-scale commercial applications are shown in Table 3.

As can be seen from the contents of Table 3, many countries and companies have conducted large-scale applications of MECoM. The three early companies that carried out large-scale applications achieved experimental results, but Luca and Ciris became bankrupt one after the other. The specific reasons for this are related to local government policies. This is also a difficulty that affects the large-scale application of MECoM. Next Fuel and Arctech developed well after that, and even cooperated with other countries and enterprises. The experience of these four companies provides valuable experience for the development of other enterprises, among which the experimental project in Yunnan, China, exists in cooperation with Next Fuel. From these results, the large-scale application of MECoM is feasible. But large-scale application implies large-scale modification of strata and landforms. Leaving aside the economic benefits, the impact of large-scale applications on the environment, the destruction of stratigraphic structure, and the pollution of groundwater must be considered.

The results of laboratory research are ultimately applied to large-scale production, which requires applications in the ground. Whether it is the placement of pretreatments, the injection of strains, the supply of nutrient solutions, or another task, the transfer of these substances needs to be carried out through well-developed fracture channels in the strata, and fracturing of the strata becomes a necessary technique.

A schematic of hydraulic fracturing is shown in Figure 6. The wellheads are divided into various types, including fracturing fluid injection ports, preform injection ports, nutrient solution injection ports, fungal solution injection ports, and air-venting ports. Diversified injection ports are easy to operate, not easily clogged, and have good stability, but the equipment requirements are high and the investment is large. So, another option can be used to merge or eliminate wellheads No. 1, No. 2, and No. 3—by giving all of the functions of these three wellheads to wellhead No. 4. The use of multi-functional wellheads allows easy operation, simplifies the equipment, and reduces the investment, but the operation is unstable, easily clogged, and not easy to maintain. Both options have advantages and disadvantages and the choice between them should be made on a case-by-case basis.

### 3.2. Potential for Biofuel Production

It is also possible to use microbial coal breakdown to produce biofuels like biodiesel and bioethanol. Enzymes generated by microorganisms have the ability to degrade the lignocellulosic components of coal, liberating simple sugars that can subsequently undergo fermentation to make biofuel. For transportation and other energy-related uses, this technique may offer a sustainable substitute for fossil fuels.

Biofuels, such as biodiesel and bioethanol, are essentially of plant origin and are obtained through controlled conversion of lignocellulosic biomass [119,120,121]. The feedstocks for biofuels are usually maize, cotton, rice straw, and sugar cane. More recently, organic wastes, oil-producing algae, and waste newspapers have also been used for biofuels [122].

As can be seen from Table 2, both microbial-enhanced coalbed methane (MECoM) technology and the production of biofuels use agricultural waste, and the microorganisms involved in the reactions overlap; so, a combination of these two technologies could be considered. Analogous to thermal power generation, which releases a large amount of harmful and toxic gasses into the atmosphere and involves the treatment of flue gasses, which requires a large amount of resources such as equipment, capital, and sites, the use of coal degradation to produce biogas instead of coal can save money to a large extent, and its clean and non-polluting characteristics are more conducive to the protection of the environment. The management of soluble organic matter in coal-seam-produced water presents a promising avenue for promoting coal biodegradation, given the presence of viable microorganisms in such environments [123]. Many wastewater treatment plants use biodegradable wastewater to produce methane for reheating [124]. It can be seen that biodegradable coalbed methane or bio-liquid fuel has great potential as a biofuel and can be used as an excellent way to help people solve their energy problems.

### 3.3. Integration with Other Renewable Energy Sources

Microbial coal degradation can also be integrated with other renewable energy sources such as solar and wind power. 

Solar energy is utilized in two main ways: power generation and heat collection. Microbial-enhanced coalbed methane (MECoM) production is affected by temperature, and in the actual use of MECoM technology, the temperature of the coal seam may not be able to meet the appropriate requirements and a large amount of heat is required to change the temperature of the coal seam. Solar collectors can be used to collect heat, and the coal seam covers a large area and the mining area is sparsely populated; so, collectors can be used on a large scale for the supply of heat from the coal seam. Electricity and heat support can be provided in the preliminary stages of increasing coalbed methane production for cultivating microorganisms, transporting, and injecting.

Some mines may be geographically located in areas with abundant wind resources, and wind power can be used to generate electricity to support increased production of coalbed methane, the daily activity of staff, and preliminary work.

Fuel cell power generation through MECoM can be a good source of energy. This is because employing direct methane oxidation and lowering the working temperature of FCs enables us to reduce a device’s size, simplify a power plant’s structure, and boost a device’s efficiency by 20–30% [125]. Microorganisms that are capable of degrading coal can be used to generate biogas, which can be used in fuel generators or other energy production systems. This integration can provide a more sustainable and efficient solution for energy production, with the potential to reduce greenhouse gas emissions and other environmental impacts.

In conclusion, MECoM has several potential applications for sustainable energy development, including in coal bioconversion for energy production, in biofuel production, and through integration with other renewable energy sources. By advancing our comprehension of the mechanisms and metabolic pathways implicated in microbial coal degradation, we can devise more proficient and successful approaches for harnessing coal reserves and fostering sustainable energy advancement.

## 4. Challenges and Limitations of Microbial Coal Degradation

Despite the potential benefits of microbial coal degradation for sustainable energy development, there are several challenges and limitations to this process. These include environmental factors affecting microbial coal degradation, as well as technical and economic challenges.

### 4.1. Environmental Factors Affecting Microbial Coal Degradation

Numerous environmental parameters, such as temperature, pH, oxygen availability, porosity, and nutrient availability, can influence how well coal is broken down by microbes [126,127,128,129,130,131]. Another important component in degradation is the limited bioavailability of coal because of its texture [132]. Therefore, changes to the coal seam environment are needed to make MECoM more effective, but coal seams are so complex and large that bringing about any one change is difficult. For instance, under the correct conditions, coal can potentially be broken down by microorganisms in reservoir water [8]. The thermophilic methanotrophic colony CBM 4 produced the most methane while it was at 60 °C, pH 7.5, and 0.1% NaCl salinity [133]. The optimal temperature for microbial coal degradation can vary depending on the microbial communities and metabolic pathways involved, and different microorganisms may have different pH requirements for optimal growth and activity. According to Bumpus et al. [128], variations in pH can modify the solubility of coal and increase the rate of deterioration. When the culture medium’s pH shifts from neutral to weakly acidic, the rate at which methane is produced rises noticeably with temperature [134]. Additionally, the functions of nutrients and trace elements in promoting microbial activity vary [135]. Methane can be held in either an adsorbed or a free state, depending on factors such as temperature, pressure, coal quality, and gas saturation [136]. Oxygen availability can also play a critical role in microbial coal degradation, as some microorganisms require oxygen for energy production while others are inhibited by it. Coal permeability, water quality, and internal scouring influence the microbial response to coal [137]. 

Depending on the strain and period of degradation, we may need to change the environmental factors, such as by increasing or decreasing the temperature of the coal seam; changing the pH value to acidic or alkaline if different microorganisms have a negative or positive correlation with pH [127]; and assessing whether oxygen is needed in the pretreatment process. Heavy elements and PAHs in coal seams can have an impact on microbial populations [32] and will require pretreatment, so we must consider how this impact can be minimized: Should biological, chemical, or physical methods be used? Fungi are more stable than bacteria [126], so should stable populations be the main focus? Adding nutrients and trace elements can improve degradation efficiency [138,139,140], but what if the nutrients needed by one period’s strain will have an inhibitory effect on the next period’s strain? Of course, there are many more questions than these. These problems may be easily solved in the laboratory, but in the practical application of MECoM technology, the difficulties will be greatly increased. In addition to in situ and ex situ experiments, to enhance the precision of biocolloid transport behavior prediction and control in porous media, a thorough examination of these physical processes and their determinants is necessary, along with the creation of relevant mathematical models and experimental techniques [141,142]. Table 4 shows degradation efficiencies and intermediate products of various microorganisms for coal mine cleanups in diverse environments and conditions.

Overall, the effectiveness of microbial coal degradation can be influenced by a range of environmental factors, and further research is needed to better understand these factors and develop more efficient and effective strategies for utilizing coal resources.

### 4.2. Efficiency, Scalability, and Economics

Key factors for the widespread application of microbial coal degradation are its scalability and efficiency. Process efficiency depends on microbial metabolism, the adaptability of the microorganisms to changing environments, and the availability of nutrients for microbial growth and activity. Different inoculum sources then influence these factors, such as native microbial communities from external sources in pristine coal seams or coal conversions [147,148,149,150], wetland sediments, cattle manure, paddy field soils [151], and termite gut [152]. On the other hand, microbial coal degradation can be a slow and complex process, requiring a variety of microorganisms, enzymes, and metabolic pathways to effectively depolymerize and degrade coal [72]. Scalability refers to the process’s capacity to increase in size without sacrificing its effectiveness. This entails a number of difficulties, including distributing nutrients and microbes uniformly over bigger reaction volumes, conducting a detailed study of a series of intermediate processes [153], preserving ideal environmental conditions throughout extended systems, and creating practical plans for gathering and handling the deteriorated coal products. Innovative technical developments and a thorough understanding of the microbial ecology and biochemistry involved in coal degradation are needed to address these issues. 

In addition to efficiency and scalability, the quality and consistency of the coal itself can also impact the effectiveness of microbial coal degradation. Coal is a porous medium, but most of the pores are small, and enlarging the porosity is also a major key to influencing the factors [154]. Coal from different sources and with different compositions can have varying levels of susceptibility to microbial degradation, which can limit the effectiveness of the process. Diverse opinions exist regarding how coal quality affects the degradation efficiency as well.

Economy is also a major obstacle to the development of MECoM. Despite the positive results of MECoM in laboratory and small-scale trials, its stability and reliability in large-scale commercial applications need to be further validated. Uncertainty in the technology may lead to long-term uncertain returns on investment, increasing the risk for investors. Investment, operation, and maintenance in large-scale applications of MECoM will increase the financial burden of enterprises. For example, the improvement of the coal seam environment, the injection of nutrient solutions, the selection and breeding of high-efficiency strains, the operation and maintenance of equipment, and the testing of the system may exceed the expectations of enterprises. Energy technology has always been the focus of attention in the world. Coalbed methane, as a kind of clean energy with limited resources, is more widely concerned by various countries, institutions, and enterprises. This has led to fierce competition for resources and increased the cost of MECoM applications. Different countries and regions have different policies related to CBM resources, which makes some enterprises afraid to invest without policy protection. In conclusion, microbial-enhanced coalbed methane production technology is economically facing many obstacles such as technological maturity and stability, costs and investment, market competition, and policy and regulatory restrictions. In order to overcome these barriers, it is necessary to continuously strengthen technological research and development, reduce costs, and improve technological awareness.

Although there are challenges and limitations in the development of MECoM production, through people’s unremitting efforts, we have achieved several breakthroughs in MECoM technology. Many difficulties have been solved for MECoM. Governments have also introduced supportive policies related to coalbed methane. This has facilitated various enterprises and researchers. Thus, it seems that the prospect of microbial-enhanced coalbed methane (MECoM) production technology is still very good, and can provide strong help for the future of the energy industry.

## 5. Future Prospects and Research Directions

### 5.1. Emerging Technologies and Strategies for Enhancing Coal Degradation Efficiency

Although there are many challenges in the application of MECoM, we can use new technologies and methods to solve them. These include creating more specialized and effective microbes for coal degradation through genetic engineering and synthetic biology, as well as creating innovative bioreactor technology to accelerate the process.

It is imperative to prioritize the development of coal biodegradation technologies in order to effectively address the challenges posed by low porosity and permeability in coal reservoirs and expedite the industrialization of coalbed methane. Biodegradation has the capacity to modify the physical properties of coal seams and enhance their solubility, permeability, and expandability [46]. One way to assess the connection or isolation of various locations is to look at the differences in methane-producing taxa among geographic basins with varying coal seams [30]. Also, the development of more effective and specialized coal-degrading microorganisms can be achieved using genetic engineering and synthetic biology. In this way, the genetic composition of microorganisms can be altered to improve their ability to break down coal or to create entirely new microorganisms with specific coal-degrading properties.

Creating innovative bioreactor technology to speed up the process is another viable strategy for improving coal degradation efficiency. Bioreactors can be used to regulate environmental parameters including temperature, pH, and oxygen availability. They are designed to offer the best conditions for microbial growth and activity.

New and more effective systems for microbial coal degradation have been developed as a result of recent developments in bioreactor technologies. Researchers have created systems that combine microorganisms capable of anaerobic respiration to produce biogas from coal, as well as bioreactors that use immobilized enzymes to increase the efficiency of coal breakdown.

China has abundant resources in its mixed layer of coal and oil, but the mining conditions are difficult and the equipment is heavy and unreliable. Safe and efficient bioreactors can lead to increased utilization of the coal–oil mixed layer. This is because oil is twice as efficient at biologically producing methane as coal [94]. 

Surfactants are a useful tool for controlling the biodegradation of coal because they dramatically alter the surface activity of bacteria, degradation products, and coal samples [90]. Biomethane can be produced by weathered coal; the more weathering, the higher the conversion potential [155]. Methane production can be increased to varied degrees by adding nutrients, including yeast extracts, peptone, glutamic acid, amino acids, vitamins, algal extracts, and others [156]. Strong degrading bacteria can be used as a pretreatment on coal to disrupt its structure and speed up the production of methane [157]. Additionally, coal is more likely to decompose with stems rather than with roots or with leaves, and coal with various kinds of straw produces higher amounts of biomethane during codegradation than either coal or straw alone [158]. The affinity of coal for methane decreases as the degree of weathering increases and the pore cleavage widens, which can enhance methane collection and utilization [155]. 

All things considered, there is potential for increasing the efficacy and sustainability of microbial coal degradation with the help of new technologies and approaches. To expand and improve these technologies and investigate novel strategies for making more sustainable and effective use of coal resources, more research is required.

### 5.2. Evaluation of Microbial Coal Degradation’s Sustainability and Effects on the Environment

Microbial coal degradation has the potential to have both positive and bad environmental effects, much like any technology that uses fossil fuels. It is crucial to take into account variables like greenhouse gas emissions, water use, and waste production in order to evaluate the process’s sustainability and influence on the environment.

As our most valuable sustainable resource, the environment is our first priority in achieving sustainable development. Although the abundance of large organic molecules in coal makes it an effective raw material for biogas production, some polycyclic aromatic hydrocarbons (PAHs) in coal also make it more difficult for other biological populations to survive, and heavy metal ions in coal can seriously contaminate the soil, with long-lasting effects on the natural ecological cycle. But all of these problems can be solved by microorganisms.

Microbial degradation of coal has the potential to reduce greenhouse gas emissions from coal mines by converting coal into biogas or other renewable fuels that can replace fossil fuels for electricity generation. However, the process itself can produce greenhouse gas emissions, especially if it relies on fossil fuels for energy or produces methane emissions in the process.

Microbial degradation of coal may require large amounts of water, especially during the microbial incubation and coal degradation steps. Microbial degradation processes can also cause some contamination of water sources. This is a problem in areas where water is scarce but minerals are abundant. However, it is feasible to isolate bacterial strains that can efficiently degrade organic pollutants and purify wastewater containing organic pollutants [159,160]. Various coal mines utilize distinct nutrient solutions and diverse microbial biomes with varying compositions to enhance the production of biogas efficiently [161]. Aromatic cyclic organic compounds (ACOs) present in coal pyrolysis wastewater (CPW) can be degraded using the lignite-activated coke-activated sludge (LAC-as) method [162,163]. 

In conclusion, microbial coal degradation is a technology that has a great impact on the environment and sustainability, but it also has some problems that we are struggling with. More investigations are needed in order to assess and improve the sustainability and environmental impact of this process.

### 5.3. Integration with Circular Economy and Waste Management

The integration of microbial coal degradation with circular economy and waste management systems presents a viable and effective means of exploiting coal resources and controlling waste streams. 

Using microbial coal degradation in circular economy systems—where waste materials are utilized as inputs for new goods and processes—is one such application. The references of this paper alone cite a large number of articles on sludge, wastewater, and waste. These articles have studied MECoM in waste treatment applications and produced good results. There are many aspects of their research, such as the use of waste coal mines for methane production, the production of other beneficial products [20], the replacement of fracking fluids [64], and the integration of agricultural wastes, which are all methods of waste management. In addition to the production of biogas described in these studies, there are many intermediate products. These intermediates can be used as raw materials in industry and agriculture. If efficient strains can be found that produce intermediates in large quantities, then the production of industrial raw materials can be taken into account while treating waste.

Integration with waste management systems, where waste materials are treated and disposed of in an environmentally appropriate way, is another possible use for microbial coal degradation. For instance, waste from coal-fired power plants can be processed via microbial coal degradation to create biogas or other beneficial products. This strategy can aid in trash reduction and encourage the management of waste from coal-fired power plants in a more sustainable manner. The previously mentioned use of agricultural waste (straw, rice straw, plant roots, etc.) for assisted yield enhancement of MECoM is feasible. Domestic wastes also have similarities with agricultural wastes, and yield enhancement experiments with domestic wastes are a feasible research direction. Of course, these wastes can be used for more than that: their use as raw materials for the production of microbial nutrient solutions and their use as materials for the cultivation of highly efficient strains of bacteria are also unnoticed technological routes. Things like industrial waste, sludge from coal mines, and wastewater are all targets for MECoM production. They can be researched accordingly in terms of pretreatment and codegradation with microorganisms, coal sludge, and other wastes. Co-bioconversion is a method that treats coal and anaerobic digestion sludge simultaneously using microbiological activity to create gas rich in methane for energy harvesting [145].

All things considered, there is potential for encouraging the development of sustainable energy and decreasing waste through the integration of microbial coal degradation with circular economy and waste management systems. To maximize and scale up these strategies for use in commercial settings, more investigation is required.

## 6. Conclusions

Coal microbial degradation in situ or ex situ is a promising method for waste management and sustainable energy development; it can increase coal utilization and reduce the cost and environmental impact of energy development. After conducting extensive research, we now know a great deal about this procedure. Nevertheless, there are still a lot of obstacles to overcome in the field of coal microbial degradation, even with our progress. Some key areas for future research and development in microbial coal degradation are as follows:(1)Making the biodegradation process clearer: Microorganisms degrade coal in complex and diverse ways, and this process requires more research and studying.(2)Building an efficient microbial activation system: It is critical to develop a system to enhance the activity and performance of the microorganisms responsible for coal degradation. This can be achieved through techniques such as genetic engineering and synthetic biology to screen for microbial strains that have efficient degradation capabilities and are highly adaptable. This could substantially increase the stability of MECoM in situ applications and is expected to provide new solutions for future energy development.(3)Creating the ideal conditions for sustainability: New technological methods are needed to improve the environmental conditions for microbial action and to accelerate the efficiency of coal bio-liquefaction and gasification. This will not only favor the development of renewable energy sources but will also give the residual coal waste great potential value.(4)Combining microbial coal degradation with the use of waste resources: Combining the use of waste resources with microbial coal degradation might improve waste management procedures and encourage a circular economy. By addressing these challenges and researching these areas, microbial coal degradation technology can be further advanced. The achievement of waste management, environmental remediation, and sustainable energy development may result from this. To maximize this procedure, investigate its economic viability, and realize the full potential of this technology for a more sustainable and environmentally friendly future, more research and development are required.(5)Generating support: In order to accomplish the goal of sustainable energy development, policymakers and the business community should simultaneously pay attention to and support this research and development.

## Figures and Tables

**Figure 1 molecules-29-03494-f001:**
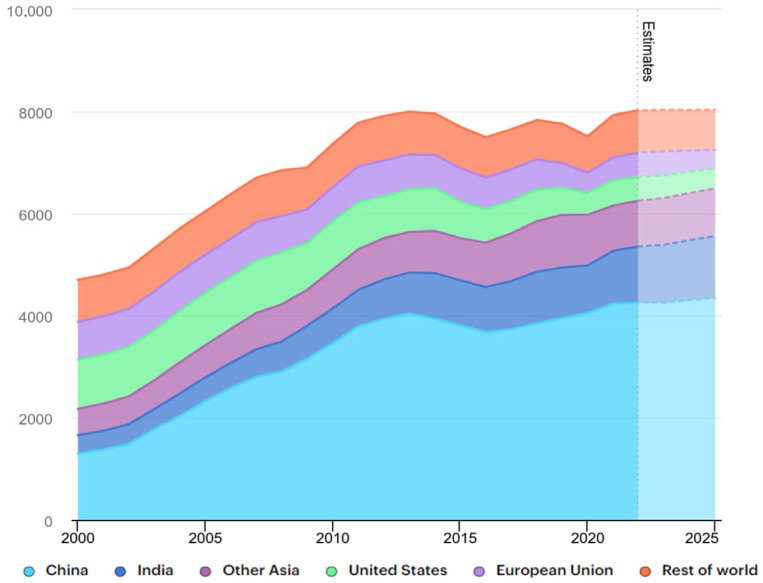
Global coal consumption, 2000–2025 (bottom to top: China, India, other Asian countries, the US, the EU, the rest of the world).

**Figure 2 molecules-29-03494-f002:**
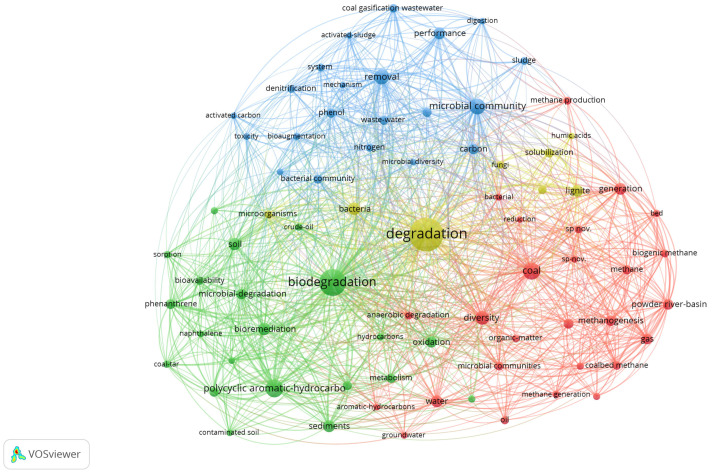
Visualization of publications involving ‘coal, degradation, microorganisms’ through keyword co-occurrence networks.

**Figure 3 molecules-29-03494-f003:**
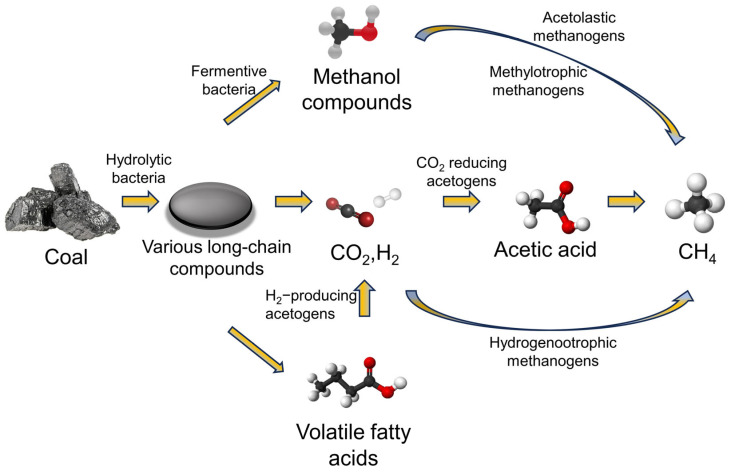
Proposed pathway for biogasifying coal to methane.

**Figure 4 molecules-29-03494-f004:**
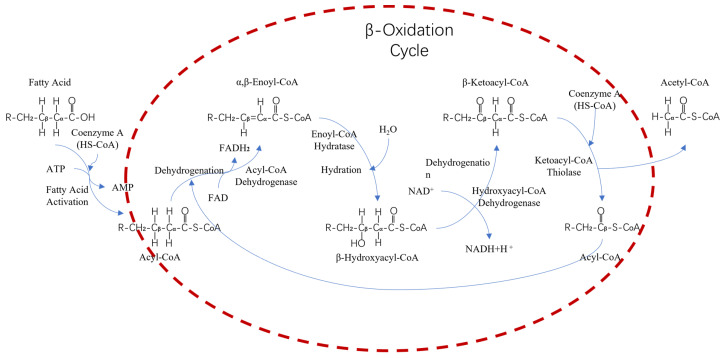
Long-chain fatty acid degradation flow chart.

**Figure 5 molecules-29-03494-f005:**
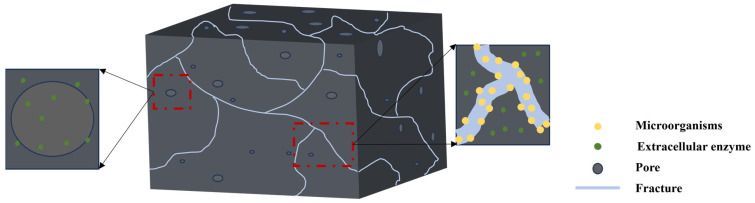
Schematic diagram of biodegradation within a coal seam fissure.

**Figure 6 molecules-29-03494-f006:**
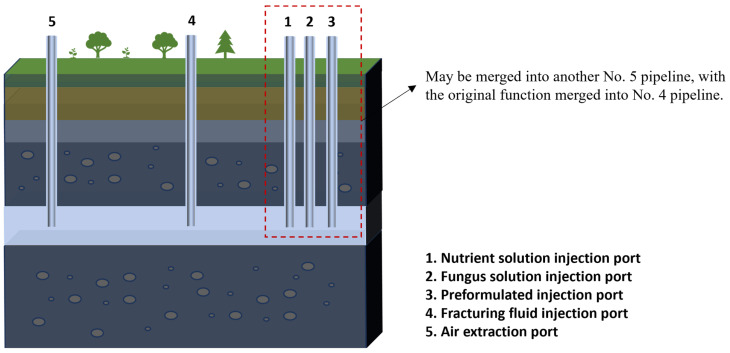
Schematic diagram of coal seam hydraulic fracturing.

**Table 1 molecules-29-03494-t001:** Microbial species and degradation mechanisms of coal.

Mechanisms of Microbial Degradation and Transformation			Types of Microorganisms		Bibliography
Mechanism of enzyme action	Peroxidase	Lignin peroxidase	Fungi	*Aspergillus* *Nematoloma frowardii* *Phanerochaete chrysosporium* *Coriolus versicolor* *Pycnoporus cinnabarinus* *Candida* *Fomes* *Edible tree fungus* *Trichoderma citrinoviride* *Polystictus* *Penicillium* *Coprinus sclerotigenis* *Stereum hirsutum*	Laborda et al. [74] Yan et al. [32] Hofrichter and Fritsche [75] Fakoussa and Frost [76] Ralph and Catcheside [77,78] Gotz and Fakoussa [79] Li et al. [64] Laborda et al. [74] Hofrichter et al. [80] Yanagi et al. [81] Feng et al. [7]
		Manganese peroxidase	Fungi	*Clitocybula dusenii* *Pleurotus ostreatus* *Nematoloma frowardii* *Collybia dryophila* *Lentinula edodes* *Fonticella*	Scheibner [82,83] Liu et al. [84] Hofrichter and Fritsche [71,80] Gotz and Fakoussa [79] Laborda et al. [74] Fakoussa and Frost [76] Steffen et al. [85]
	Phenoloxidase	Laccase	Fungi	*Trichoderma* *Aspergillus* *Trichoderma citrinoviride* *Lentinula edodes* *Nematoloma frowardii* *Polyporus versicolor* *Poria monticola* *Coprinus sclerotigenis* *Alternaria* *Polystictus consors (Berk.) Teng* *Coriolus hirsutus(Wulf: Fr.) Quel* *Bjerkandera adusta (Willd.: Fr.) Karst R59* *Azoarcus* *Paecilomyces Bain* *Coriolus versicolor*	Fakoussa and Frost [76] Gotz and Fakoussa [79] Hofrichter and Fritsche [83] Cohen and Gabriele. [11] Holker et al. [86] Hofrichter et al. [80] Yan et al. [32] Yanagi et al. [87] Yanagi et al. [81] Feng et al. [7] Belcarz et al. [88] Fu et al. [54]
			Bacteria	*Bacillus licheniformis* *Alicycliphilus* *Pseudomonas adaceae* *Polaromonas* *Geobacter* *Mycobacterium* *Sphingomonas* sp.	Wu et al. [4] Ding et al. [89] Fu et al. [54] Shen et al. [90]
	Hydrolysis enzyme	Lipase	Fungi	*Trichoderma longibrachiatum Rifai* *Karst R59* *Mortierella* sp. *Bjerkandera adusta (Willd.: Fr.)* *Aspergillus* *Fusarium oxysporum Schltdl.*	Holker et al. [86] Belcarz et al. [88] Yan et al. [32]
			Bacteria	*Pseudomonas cepacia 122* *Pseudomonas cepacia AC100* *Pseudomonas cepacia ATCC 21808* *Acidovorax* *Sedimentibacter* *Proteobacteria* *Enterobacter* *Betaproteobacteria* *Deltaproteobacteria* *Clostridium* *Pseudomonas cepacia DLC-07* *Alphaproteobacteria* *Gammaproteobacteria*	Gupta et al. [91] Kilbane et al. [92] Kordel et al. [93] Li et al. [94] Li et al. [66] Fu et al. [54]
			Proteobacteria	*Desulfovibrio* *Desulfobacterota* *Geobacter* *Desulfococcus oleovorans Hxd3* *Syntrophobacter* *Syntrophomonas*	Li et al. [64] Fu et al. [54] Campbell et al. [95]
Mechanism of alkali dissolution			Bacteria	*Bacillus cereus* *Bacillus pumilus* *Bacillus subtilis* *Pseudomonas putida*	Maka et al. [65] Machnikowska et al. [96]
			Actinomycetes	*Streptomyces badius.* *Streptomyces viridosporus*	Quigley et al. [97] Wu et al. [4]
			Fungi	*Fusarium oxysporum Schltdl.* *Trichoderma longibrachiatum Rifai*	Holker et al. [86]
		Reductase	Bacteria	*Campilobacterota* *Firmicutes* *Acetobacterium* *Smithella*	Fu et al. [54]
Surfactant mechanism of action			Actinomycetes	*Streptomyces viridosporus* *Streptomyces flavovirens*	Wu et al. [4]
			Fungi	*Neosartorya fischeri*	Lgbinigie. [98]
Mechanism of action of chelating agents			Fungi	*Trichoderma longibrachiatum Rifai* *Fusarium oxysporum Schltdl.*	Holker et al. [86]
Methoxydotrophic mechanism of action			Bacteria	*Acetoclastic Methanosarcina* *Methanobacteriaes* *Candidatus Methanothrix Paradoxum* *Methanofastidiosa* *Methermicoccus shengliensis* *Euryarchaeota* *Thermovirga* *Clostridiales* *Methanomicrobiales* *Methermicoccus shengliensis AmaM* *Methermicoccus shengliensis ZC-1*	Mayumi et al. [99] Li et al. [64] Li et al. [66] Fu et al. [54]

**Table 2 molecules-29-03494-t002:** Effect of treatments on microbial methane production.

Treatment	Materials	Effect	Bibliography
In situ nutritional modification	Rice straw	Methane yield of 684.83 µmol/g coal;	Li et al. [36] Guo et al. [106] Guo et al. [107]
	Sweet sorghum straw	Methane yield of 612.98 µmol/g coal;	
	Wheat straw	Methane yield of 537.31 µmol/g coal;	
	Corn straw	Methane yield of 46.95 µmol/g coal;	
	Rice straw	Methane yield of 93.65 µmol/g coal; Plant roots, stems, and leaves, corresponding to different qualities of coal, can produce different effects.	
Enrichment culture	Nitrogen amendment	Increased 1.89 to 3.43 times;	Li et al. [108] Kurnani et al. [109]
	Rumen liquid from beef cattle	The powdered rumen in 10^−7^ dilution can be used to increase methane production in lignite, subbituminous, and bituminous reserves.	
Physical fracturing	Hydraulic fracturing	Significant increase in degradation strains.	Li et al. [66] Robbins et al. [108]
Biological/non-biological pretreatment	Aerobic fungi or bacteria	Bacterial pretreatment products mainly include single-ring aromatics, long-chain alkanes, and long-chain fatty acids. Fungal pretreatments were predominantly identified as polyaromatic hydrocarbons, single-ring aromatics, aromatic nitrogen compounds, and some aliphatics.	Liu et al. [16] Haide et al. [104] Haide et al. [110] Chen et al. [111] Xia et al. [112]
	H_2_O_2_	Methane production of 529.3 µmol/g;	
		The methane production of the sub-bituminous coal PEN9-003 increased up to 10 times to 223.7 μmol/g;	
	White-rot fungi	Hydrogen production was 1.32 mL/g and methane production was 5.78 mL/g.	

**Table 3 molecules-29-03494-t003:** A survey of practical use cases of MECoM.

Time	Mechanism	Site	Projects and Effects	Bibliography
2006	Luca Technologies	Powder River Basin of northeastern Wyoming and southeastern Montana in the western USA	Methane production increased in 58 wells, with an efficiency rate of 22%.	Ritter et al. [113]
2012	Ciris	Powder River Basin of northeastern Wyoming and southeastern Montana in the western USA; Headquartered in Centennial, Colorado.	Nutrient infusion was carried out and, after a long period of testing, an increase in yield was observed.	Ritter et al. [113]
2013	Next FUEL	Powder River Basin of northeastern Wyoming and southeastern Montana in the western USA; Headquartered in Sheridan, Wyoming	Nutrient introduction was mainly carried out.	Ritter et al. [113]
2013	North China Oilfield	It is located in the North China Oilfield area in Hebei Province, China.	After 1 year, 8 orthogonal experimental designs, and more than 200 sampling tests, the gas content of microbial methane was increased from the initial 4.1% to 97.8%.	North China Oilfield [114]
2013	Yunnan Provincial Energy Investment Group Co., Ltd. and Yunnan Baocheng New Energy Co., Ltd. in collaboration with Next Fuel Inc.	Located in Huaning County, Yuxi City, Yunnan Province, China.	A commercialization project using Biological Coalbed Methane Technology (BCTG) was carried out and achieved remarkable results.	Yunnan Provincial Energy Investment Group Co., Ltd. and Yunnan Baocheng New Energy Co., Ltd. in collaboration with Next Fuel Inc. [115]
2019	Daqing Oilfield	It is located in the northern part of the Songnen Plain in Heilongjiang Province and Daqing City, China.	It was found that when using microbial oil drive, the annual gas production per gram of crude oil can be more than 150 mL, and the oil and gas conversion rate is more than 10%.	Daqing Oilfield [116]
2023	Arctech	Located in Centerville, Virginia, USA	ARCTECH developed MicGASTM technology by adapting wood termites to eat coal and then using the microbes isolated from their guts to digest coals in the presence of appropriate nutrient components. This technology can be applied to low-cost installations in wastewater treatment plants. This technology has also been used to convert residual oil from unminable coal, shale, and reservoirs into clean methane gas. The solid residues from anaerobic treatment are also not waste, but are rich in organic humus.	Arctech [117]
2023	Xinjiang Kelinside New Energy Co., Ltd.	It is located in Fukang City, Changji Hui Autonomous Prefecture, Xinjiang Uygur Autonomous Region, China.	Fracture testing was formally initiated on the FK18-2L horizontal well and continued for 15 days.	Xinjiang Kelinside New Energy Co., Ltd. [118]

**Table 4 molecules-29-03494-t004:** Yield and intermediates of different microorganisms in different environments and raw materials.

Microbiology	Microbial Source	Coal Type	T/°C	pH	Methane Yield	Metabolite	Bibliography
Nocardia mangyaensis (N. mangyaensis; CICC11046)	The China Center of Industrial Culture Collection (CICC)	Fresh coal samples	30 °C	—	The biodegradation rate was 65.2%.	Phenol; alcohol; ether; ester	Shi et al. [143]
Bacillus licheniformis (B. licheniformis; CICC10092)		Fresh coal samples	30 °C	—	The biodegradation rate was 58.5%.		
Nocardia mangyaensis (N. mangyaensis; CICC11046) and Bacillus licheniformis (B. licheniformis; CICC10092)		Fresh coal samples	30 °C	—	The biodegradation rate for degradation order N→B was 82.1%.		
		Fresh coal samples	30 °C	—	The biodegradation rate for degradation order B→N was 75.5%.		
		Fresh coal samples	30 °C	—	The biodegradation rate of the two microorganisms together was 48.3%.		
Petrimonas, Lysinibacillus, Proteiniphilum, Bacillus, Cloacibacillus, Methanomassiliicoccus, and Methanosarcina.	Huainan	Fresh coal samples	—	6.8	The cumulative methane yield was 4.521 mL/g.	Alkanes; aromatics; amines; unsaturated fatty acids; esters	Su et al. [144]
	Hebi	Fresh coal samples	—	6.8	The cumulative methane yield was 3.151 mL/g.		
	Zhaogu	Fresh coal samples	—	6.8	The cumulative methane yield was 2.013 mL/g.		
—	The Ordos Basin	Fresh coal samples	60 °C	—	The majority of the products were alkanes, with concentrations ranging from 64.2 to 220.6 ng/L.	Aliphatic hydrocarbons; polycyclic aromatic hydrocarbons; heterocyclic phenols; esters; ethers; alcohols; other aromatic compounds	Bao et al. [123]
—	Domesticated mixed fermentation microorganisms	Fresh coal samples	35 °C	—	Gas production was not examined and pore changes (transformation of micropores into transition and mesopores) were detected.	—	Li et al. [22]
Methanobacteriales, Methanocellales, Methanococcales, Methanobacteriales, Methanomicrobiales, and Methanosarcinales.	Huangling Coal Mine at Shaanxi Huangling Mining Co., Ltd., China.	Fresh coal samples	37 °C	—	By day 90, the methane yield was 1.06 µmol/(g·d).	Benzenoids (108 metabolites); organoheterocyclic compounds (125 metabolites); phenylpropanoids and polyketides (48 metabolites); lipids and lipid-like molecules (48 metabolites); fatty acyls (16 metabolites), etc.;	Li et al. [94]
Methanobacterium, Methanobrevibacter, Methanoculleus, and Methanosarcina.		Crude oil	37 °C	—	By day 90, the methane yield was 2.29 µmol/(g·d).	82 organoheterocyclic compounds; 58 benzenoids; 46 organic acids and derivatives; 18 fatty acyls	
—	The deep mine water of the Guhanshan Mine in Jiaozuo City, Henan Province.	Low-rank lignite	35 °C	7.0 ± 0.05	The cumulative methane yield was 152.11 µmol/g.	—	Xia et al. [46]
	The deep mine water of the Guhanshan Mine in Jiaozuo City, Henan Province	Medium-rank bituminous coal	35 °C	7.0 ± 0.05	The cumulative methane yield was 80.57 µmol/g.	—	
Mortieralla, Cladosporium, Alternaria, Cladosporium, Fusarium, Aspergillus, and Methanosarcina.	Qinshui Basin	Lignite	35 °C	—	The maximum methane production was 6578.51 μmol.	Fatty acids; amino acids; nitrogenous compounds; alcohols; aromatic acid	Yan et al. [32]
Methanosarcina, Methanobacterium, Methanomassiliicoccus, Methanothrix, and Methanoculleus.	Low-volatility anthracite (Sihe No. 2 Coal Mine)	High-volatility bituminous coal	30 °C	Fell to 6.11 ± 0.1 during days 1–5; increased to 8.11 ± 0.3 on days 5–40.	The cumulative methane production rate was 207.3 μmol/g.	Heterocyclics; benzenoids; aliphatic acids; polymers (mass charge ratio >400)	Liu et al. [84]
	High-volatility bituminous coal (Panji No. 3 Coal Mine)	High-volatility bituminous coal	30 °C	Fell to 6.11 ± 0.1 during days 1–5; increased to 8.25 ± 0.2 on days 5–40.	The cumulative methane production rate was 243.3 μmol/g.		
	Medium-volatility coking coal (Pinggou Coal Mine)	High-volatility bituminous coal	30 °C	Fell to 6.11 ± 0.1 during days 1–5; increased to 7.76 ± 0.2 on days 5–40.	The cumulative methane production rate was 163.1 μmol/g.		
—	An active mine site in the U.S.	Lignite and subbituminous	35 °C	Around 7.2	The cumulative amounts of produced biogas at day 40 were 62.3 mL/g sludge, 62.8 mL/g sludge, and 67.1 mL/g sludge, respectively; these values eventually reached 120.9 mL per gram of sludge (mL/g sludge), 144.4 mL/g sludge, and 161.3 mL/g sludge in blank, subbituminous, and lignite, respectively.	—	Rahimi et al. [145]
—	Guhanshan Mine, Jiaozuo, Henan, China.	Long-flame coal	35 °C	—	The methane production rate was 53.6%, and cumulative methane production was 4.28 mL/g.	Monosaccharides; different amino acids; large amounts of fatty acids and glycerol	Xia et al. [146]

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
