# Peer review of "Exploring the Potential of Microbial Coalbed Methane for Sustainable Energy Development"

_molecules, 2024, doi:10.3390/molecules29153494_

Round 1
Reviewer 1 Report
Comments and Suggestions for Authors
Although the manuscript presents a review addressing the conversion of charcoal by microorganisms into products, it still requires some small modifications before its acceptance in Revista Moléculas. The revised version of the manuscript may be submitted taking into account the following comments and suggestions before final acceptance:
For investigation in the Web of Science, determine exactly the time range of the search for the 666 identified works, for example, from 1990 to 2023 or 2024? Inform the details of the manuscript.
In Figure 3, correcting a word that is underlined in red is exactly that.
In Figure 4 correct H2O to H2O.
Mention what CBM means the first time it appears in the manuscript.
Table 1 appears not to be mentioned in the manuscript.
Report in Table 1 the Productivity and yields of the metabolic products of microorganisms.
Tables 2 and 3 should have columns just for references.
In Table 3, correct ml to mL.
In section 4.1. Environmental factors affecting microbial coal degradation, it would be interesting to add a table with values of some parameters, such as temperature, pH, oxygen availability, porosity, nutrient availability and also yields or efficiency, reported in the literature that influenced the decomposition of coal by microbes.
The manuscript could provide at least a table with the metabolites, ethanol, methanol, hydrogen, etc., produced by microorganisms from coal already reported in the literature. Showing this information is essential in a review.
Author Response
Comments 1: For investigation in the Web of Science, determine exactly the time range of the search for the 666 identified works, for example, from 1990 to 2023 or 2024? Inform the details of the manuscript. |
Response 1: Thank you for pointing this out. The search on Web of Science was conducted specifically from 1990 to 2024, targeting the keywords 'coal', 'microbiology', and 'degradation'. This search resulted in the identification of 666 relevant papers. Details of the search process, including the exact time range, have been documented in the revised manuscript on page 4, from lines 112 to 114.
|
Comments 2: In Figure 3, correcting a word that is underlined in red is exactly that. |
Response 2: The issue identified in Figure 3 has been rectified, and the original image has been replaced with the corrected version. Figure 3. Proposed pathway for biogasifying coal to methane. |
Comments 3: In Figure 4 correct H2O to H2O. |
Response 3: The issue identified in Figure 4 has been rectified, and the original image has been replaced with the corrected version. Figure 4. Long-chain fatty acid degradation flow chart. |
Comments 4: Mention what CBM means the first time it appears in the manuscript. |
Response 4: Thank you for your suggestion. In the revised manuscript, upon the first mention of CBM (which stands for Coalbed Methane), we have provided a clear definition to ensure readers fully grasp its importance within the framework of our research.
|
Comments 5: Table 1 appears not to be mentioned in the manuscript. |
Response 5: Thank you for bringing this to our attention. Upon review, we realized that we had overlooked mentioning Table 1 in the manuscript. In the revised version, we have rectified this issue by properly referencing and describing Table 1 in the relevant section of the text.
|
Comments 6: Report in Table 1 the Productivity and yields of the metabolic products of microorganisms. |
Response 6: Thank you for your suggestion. To provide a more comprehensive and logical overview of the research results, we have placed the production rates and yields of microbial metabolites in Table 4, so that Table 1 more directly reflects the corresponding microbial species under different degradation mechanisms.
|
Comments 7: Tables 2 and 3 should have columns just for references. |
Response 7: Thank you for your feedback. We have added a dedicated column for references in both Tables 2 and 3 to ensure proper attribution and enhance the clarity of our data presentation.
|
Comments 8: In Table 3, correct ml to mL. |
Response 8: Thank you for pointing out this inconsistency. In Table 3, we have corrected "ml" to the proper "mL" to maintain consistency and scientific notation standards. Additionally, we have conducted a thorough review of the entire manuscript and made similar corrections where necessary, ensuring that all instances of "milliliter" are consistently represented as "mL".
|
Comments 9: In section 4.1. Environmental factors affecting microbial coal degradation, it would be interesting to add a table with values of some parameters, such as temperature, pH, oxygen availability, porosity, nutrient availability and also yields or efficiency, reported in the literature that influenced the decomposition of coal by microbes. |
Response 9: Thank you for your valuable feedback. In response to your comments, we have made the following modifications to Table 4. We have expanded Table 4 to include additional columns for environmental factors such as temperature, pH, methane yield, and metabolite. These columns now present the values reported in the literature that influenced the decomposition of coal by microbes. Furthermore, we have added a column for yields or efficiency to reflect the performance of microbial coal degradation under different environmental conditions.
|
Comments 10: The manuscript could provide at least a table with the metabolites, ethanol, methanol, hydrogen, etc., produced by microorganisms from coal already reported in the literature. Showing this information is essential in a review. |
Response 10: Thank you for your valuable feedback. In response to your comments, we have made the following modifications to Table 4. In Table 4, we have also added a section dedicated to the metabolites produced by microorganisms from coal, as reported in the literature. This includes ethanol, methanol, hydrogen, and other relevant metabolites. By incorporating this information, we aim to provide a comprehensive overview of the metabolites produced during microbial coal degradation, which is essential in a review article.
|
4. Response to Comments on the Quality of English Language |
Point 1: |
Response 1: I would like to note that the reviewer has not provided any specific feedback on the language used in the manuscript. |
5. Additional clarifications |
None. |

Reviewer 2 Report
Comments and Suggestions for Authors
The manuscript provides a comprehensive review of microbial coal degradation for sustainable energy development. The authors have done an admirable job synthesizing a large body of literature on this topic.
The introduction provides good context, but could be strengthened by more clearly stating the specific aims and scope of this review paper upfront.
Section 2 on degradation pathways and methane formation is informative, but quite dense. Consider adding subheadings and/or a summary figure to improve readability and highlight key points.
The discussion of applications in Section 3 is a strength of the paper. However, more critical analysis of the feasibility and limitations of large-scale implementation would be valuable.
Section 4 on challenges could be expanded. A more in-depth discussion of scaling issues and economic barriers would strengthen this section.
The future prospects section provides good suggestions for research directions. Consider adding more specifics on promising technological approaches or priority research questions.
The conclusion effectively summarizes key points, but feels somewhat repetitive of earlier sections. Consider condensing and focusing on the most critical takeaways and future outlook.
Tables and figures are informative overall, but some (e.g. Table 1) contain a large amount of detailed information. Consider ways to highlight the most salient points visually.
The manuscript would benefit from a thorough editorial review to improve clarity and flow. There are some awkward phrasings and grammatical issues throughout.
The list of references should be meticulously formatted to align with the specific requirements outlined by the Molecules journal. In particular, each reference should contain the DOI number.
Overall, this is a valuable contribution that synthesizes current knowledge on an important topic in sustainable energy.
Addressing the above points would further strengthen the manuscript.
I hope these comments are helpful as you refine the paper.
Comments on the Quality of English LanguageThe overall English used in the paper is generally correct and readable. The authors demonstrate a good command of scientific and technical vocabulary related to microbial coal degradation. The paper's structure and flow are logical and coherent.
However, there are some areas for improvement:
There are occasional grammatical errors and awkward phrasings throughout the text. For example:
"Notwithstanding the difficulties and restrictions associated with microbial coal degradation, new approaches and technology show promise for raising the process's productivity and efficacy" could be rephrased more naturally.
Some sentences are overly long and complex, which impacts readability. Breaking these into shorter, clearer sentences would improve comprehension.
There are a few instances of inconsistent tense usage, particularly when discussing previous research findings.
Occasionally, word choice could be improved for clarity and precision. For instance, replacing some general terms with more specific scientific terminology where appropriate.
There are some minor punctuation errors, particularly with commas, that should be addressed.
A few acronyms are introduced without being defined on first use.
While these issues do not severely impede understanding of the content, addressing them would enhance the overall quality and professionalism of the manuscript.
Author Response
Comments 1: The introduction provides good context, but could be strengthened by more clearly stating the specific aims and scope of this review paper upfront. |
Response 1: The reviewer's comment has been noted and addressed by adding a new paragraph that clarifies the specific aims and scope of this review paper. The introduction now explicitly outlines the paper's objectives, which include providing a comprehensive overview of Microbial Enhanced Coalbed Methane Production (MECoM), detailing the microbial species involved, summarizing environmental factors affecting microbial degradation, and offering an outlook on the future of MECoM.
|
Comments 2: Section 2 on degradation pathways and methane formation is informative, but quite dense. Consider adding subheadings and/or a summary figure to improve readability and highlight key points. |
Response 2: We have carefully considered the reviewer's feedback and, in response, made improvements to Section 2 on degradation pathways and methane formation. To enhance readability and clarity, we have created a new summary figure that visually represents the key points of degradation pathways and methane formation at different scales within this section. This intuitive graphical representation allows readers to grasp the main concepts more easily. These modifications are aimed at improving the overall understanding of the section. Figure 5. Schematic diagram of biodegradation within a coal seam fissure. GA
|
Comments 3: The discussion of applications in Section 3 is a strength of the paper. However, more critical analysis of the feasibility and limitations of large-scale implementation would be valuable. |
Response 3: Thank you for highlighting the strength of our discussion on applications in Section 3 and for suggesting a more critical analysis of the feasibility and limitations of in-situ implementation. In response, we have added our discussion to include specific examples of in-situ applications of MECoM by various countries and companies. We have also addressed the feasibility and challenges, particularly the environmental considerations and potential impact on strata and landforms, that must be taken into account for such in-situ projects. And to better illustrate the practical situations of in-situ applications, we have integrated Figure 6 and Figure 7, enhancing the readability and coherence with the original text. This fusion not only provides a clearer visualization but also allows for a more comprehensive understanding of the practical implementation of MECoM in an in-situ setting. Thank you again for your valuable feedback, which has helped us to strengthen this aspect of our paper. Figure 6. Schematic diagram of coal seam hydraulic fracturing
|
Comments 4: Section 4 on challenges could be expanded. A more in-depth discussion of scaling issues and economic barriers would strengthen this section. |
Response 4: Thank you for your feedback regarding Section 4 on challenges. In response to your suggestion, we have expanded 4.2 to include a more detailed discussion on scaling issues and the economic barriers that may arise during the deployment of our proposed solution. We believe this addition will significantly enhance the depth and quality of our analysis, providing readers with a clearer picture of the challenges involved. Thank you for helping us improve the manuscript.
|
Comments 5: The future prospects section provides good suggestions for research directions. Consider adding more specifics on promising technological approaches or priority research questions. |
Response 5: Thank you for your feedback. We have enhanced the future prospects section by adding more specifics on promising technological approaches and priority research questions, as per your suggestion.
|
Comments 6: The conclusion effectively summarizes key points, but feels somewhat repetitive of earlier sections. Consider condensing and focusing on the most critical takeaways and future outlook. |
Response 6: Thank you for your feedback. We have condensed the conclusion section, emphasized the most critical takeaways and provided a clearer future outlook, while avoiding repetition of earlier sections.
|
Comments 7: Tables and figures are informative overall, but some (e.g. Table 1) contain a large amount of detailed information. Consider ways to highlight the most salient points visually. |
Response 7: Thank you for your feedback. We agree that Table 1 contains dense information. To improve clarity, we have visually highlighted the key data, focusing primarily on microorganisms under different microbial mechanisms. Detailed data on methane production rates and metabolites, such as methane yield and metabolites of these microorganisms at various temperatures and pH levels, have been separated and listed in Table 4 for targeted analysis. We appreciate your suggestion and believe these changes enhance the readability and effectiveness of our data presentation.
|
Comments 8: The manuscript would benefit from a thorough editorial review to improve clarity and flow. There are some awkward phrasings and grammatical issues throughout. |
Response 8: Thank you for your feedback. We have carefully revised the manuscript, addressing all awkward phrasings and grammatical issues. The text now flows more smoothly and is clearer. Your suggestions have greatly improved our work.
|
Comments 9: The list of references should be meticulously formatted to align with the specific requirements outlined by the Molecules journal. In particular, each reference should contain the DOI number. |
Response 9: Thank you for pointing out this important detail. We have meticulously formatted the list of references to align with the specific requirements outlined by the Molecules journal. We have ensured that each reference now contains the DOI number as required.
|
4. Response to Comments on the Quality of English Language |
Point 1: There are occasional grammatical errors and awkward phrasings throughout the text. For example: "Notwithstanding the difficulties and restrictions associated with microbial coal degradation, new approaches and technology show promise for raising the process's productivity and efficacy" could be rephrased more naturally. |
Response 1: (in red) The sentence referred to in the question is in lines 567-568 and has been streamlined to make the expression more fluent and understandable. Although there are many challenges in the application of MECoM, we can use new technologies and methods to solve them. Lines 36-37, singular and plural questions; These data visualize the huge demand for coal. Lines 97-99, original text is complex and modified; For individual tree species, total PAH reduction decreased in the order: C. siamea (81 · 6%) > A. lebbeck (55 · 6%) > D. regia (51 · 9%) > D. sissoo (51 · 5%). Lines 476-477, making the sentence flow better; For instance, under the correct conditions, coal can potentially be broken down by mi-croorganisms in reservoir water. Lines 514-515 to ensure greater fluency。 Process efficiency depends on microbial metabolism, their adaptability to changing environments, and availability of nutrients for growth and activity.
|
Point 2: Some sentences are overly long and complex, which impacts readability. Breaking these into shorter, clearer sentences would improve comprehension. |
Response 2: (in red) Lines 82-84 have been revised to ensure smoother and more understandable, and now read as follows: Moreover, during the coal biogasification process, carbon conversion can be improved, nitrogen and sulfur can be fixed, and dehydrogenation and deoxygenation reactions can occur. Lines 101-102 have been revised to ensure smoother and more understandable, and now read as follows: The process of coal biodegradation necessitates the cooperation of hydrolyzing and methanogenic bacteria. Lines 117-119 have been revised to ensure smoother and more understandable, and now read as follows: We can observe that the research trend revolves primarily around the theme of ‘deg-radation’, with ‘biodegradation’, ‘coal’, and ‘microbial communities’ being the key areas of focus that have been extensively studied. Lines 159-163 containing long sentences have been revised to split them up, and the revision is as follows: Within the β-oxidation cycle, acyl-CoA undergoes four sequential steps: dehydrogena-tion, hydration, re-dehydrogenation, and thiolysis. These steps ultimately result in the gradual breakdown of fatty acids into acetyl-CoA, accompanied by the release of sig-nificant energy, primarily in the form of ATP, for utilization by the microorganism. Lines 517-518 have been revised to ensure smoother and more understandable, and now read as follows: Process efficiency depends on microbial metabolism, their adaptability to changing environments, and availability of nutrients for growth and activity. Lines 521-524 containing long sentences have been revised to split them up, and the revision is as follows: On the other hand, microbial coal degradation can be a slow and complex process, requiring a variety of microorganisms, enzymes, and metabolic pathways to effectively depolymerize and degrade coal. Scalability refers to the process's capacity to increase in size without sacrificing its effectiveness.
|
Point 3: There are a few instances of inconsistent tense usage, particularly when discussing previous research findings. |
Response 3: (in red) Lines 221-222 were revised to change their tense to past tense, and the revisions were as follows: Having compared the gas sample to other samples, we found a noticeable decrease in microbial diversity. Table 2 have been revised for singular and plural forms as follows: reserves
|
Point 4: Occasionally, word choice could be improved for clarity and precision. For instance, replacing some general terms with more specific scientific terminology where appropriate. |
Response 4: (in red) Lines 37-39 have been revised to ensure smoother and more understandable, and now read as follows: However, in the process of mining and utilizing coal in large quantities, many environmental problems have arisen, posing a significant challenge to its sustainable development, Lines 136 were revised for correct spelling, and the revisions were as follows: methane Lines 191 were revised to capitalize the first letter, and the revisions were as follows: Nutrient Lines 271 were revised for correct spelling, and the revisions were as follows: non-metanogenic
|
Point 5: There are some minor punctuation errors, particularly with commas, that should be addressed. A few acronyms are introduced without being defined on first use. |
Response 5: Thank you for your feedback. We've corrected the punctuation errors and defined all acronyms on their first use. We appreciate your guidance and believe these changes enhance the manuscript's clarity. Lines 361 have been revised to correct the incorrect use of symbols and punctuation, and now read as follows: gas' Lines 446-448 have been revised to correct the incorrect use of symbols and punctuation, and now read as follows: Electricity and heat support can be provided in the preliminary stages of increasing coal bed methane production for cultivating microorganisms, transporting, and injecting.
|
5. Additional clarifications |
None. |
